**Data Availability Statement:** All relevant data are within the manuscript and its Supporting Information files.

**Funding:** The sources of funding for our study were Grants-in-Aid for Scientific Research (KAKENHI) of the Japan Society for the Promotion

# Development of a social contact self-efficacy scale for 'third agers' in Japan

**Moemi Oki** [ID] [1☯], **Etsuko Tadaka** [ID] [2☯]*

1 Department of Community Health Nursing, Graduate School of Medicine, Yokohama City University, Yokohama, Kanagawa, Japan, 2 Department of Community and Public Health Nursing, Faculty of Medicine/Graduate School of Health Sciences, Hokkaido University, Sapporo, Hokkaido, Japan

☯ These authors contributed equally to this work.
* e_tadaka@pop.med.hokudai.ac.jp

## Abstract

### Background

"Third agers" are people over retirement age in relatively good health; third agers make up an increasing percentage of the global population as the world's longevity increases. Therefore, the challenge of prolonging a healthy third age and shortening the unhealthy period during the "fourth age" in the global health and social contexts is important in this process. However, no means to measure and support this has been developed as yet. We developed the Social Contact Self-Efficacy Scale for Third Agers (SET) and evaluated its reliability and validity.

### Methods

We used a self-administered mail survey covering 2,600 randomly selected independent older adults living in Yokohama, Japan. The construct validity of the SET was determined using exploratory factor and confirmatory factor analyses. Its criterion-related validity was assessed using the General Self-Efficacy Scale (GSES), the Japan Science and Technology Agency Index of Competence (JST-IC), and subjective health status.

### Results

In total, 1,139 older adults provided responses. Exploratory and confirmatory factor analyses identified eight items within two factors: social space mobility and social support relationship. The final model had a Cronbach's alpha 0.834, goodness-of-fit index 0.976, adjusted goodness-of-fit index 0.955, comparative fit index 0.982, and root mean square error of approximation 0.050. There was good correlation between scale scores and the GSES (r = 0.552, p < 0.001), JST-IC (r = 0.495, p < 0.001) and subjective health status (r = 0.361, p < 0.001).

### Conclusions

The SET showed sufficient reliability and validity to assess self-efficacy in promoting social contact among third agers. This scale may help third agers in gaining and expanding

of Science (PI: Dr. Etsuko TADAKA). The funders
had no role in study design, data collection and
analysis, decision to publish, or preparation of the
manuscript. No authors received a salary or fees
from any of the funders.

**Competing interests:** The authors have declared
that no competing interests exist.

opportunities for social contact, which can improve their physical health and quality of life
and contribute to care prevention and healthy longevity.

## Introduction

The concept of life stages has altered dramatically in the 21st century, as people live increasingly longer. Laslett (1996) posited a new framework for age-independent life stages, consisting of four functional periods or 'ages'. The first age is the age of dependence and socialization; the second is the age of independence and reproduction, family and responsibility to society, the third is the age of achievement and the fourth is the age of dependence, senility and death [1]. The fourth age is characterized by functional decline and increased dependency. In contrast, the third age, which starts with retirement, is a period of relatively good health with the potential for active social engagement forming a solid base for healthy ageing. In developed countries around the world, the proportion of 'third agers' in the population is increasing. The great challenge for third agers is to prolong this period healthily and therefore shorten the unhealthy period of the fourth age as much as possible [2, 3]. As of 2020, the proportion of older people in Japan's population was 28.8% [4], the highest in the world. By 2030, one in every three people will be 65 years old or more, and one in five 75 or older. Japanese life expectancy is among the highest in the world and it keeps on increasing. Along with the extension of life span, healthy life expectancy—number of years people are expected to live in good health, that is, without needing long-term care and support—is also increasing. It was 74.79 years old for women, and 72.14 years old for men in 2020 [5]. However, the gap between life expectancy (87.32 years old in women and 81.25 years old in men) and healthy life expectancy has remained the same, 9.1 years in men and 12.5 years in women.

Previous research has shown that one key to health and longevity among third agers is their relationships with society [6]. Especially after retirement, with accompanying changes in their environment and relationships, individuals often face challenges such as having to remain homebound, declining ability to perform activities of daily living (ADLs), social isolation, decline in quality of life, and increased risk of death [7]. In contrast, socially active older adults have less cognitive decline [8, 9] and are at lower risk of transitioning to needing long-term care and support [10]. Moreover, maintaining social contact has long been known to be related to health and well-being in old age [11], and positive effects of social relationships among third agers have been reported in several recent studies [12–14].

There are two elements in social relationships that contribute to improved health and longevity among third agers, namely, "social space mobility" and "social support relationships". "Social space mobility" is defined as extent of individuals' social contacts, as related to both living space and mobility. "Social support relationships" is defined as interactions with individuals' social contacts as well as the ability to build and maintain new social relationships (including with friends, neighbors, public and private support agencies, and primary care physicians) while preserving existing relationships in one's retired life. Enhanced social participation among older people means maintaining wide social space mobility and enhanced social support relationships. Previous studies have shown that older adults who maintain a wider range of activities have improved physical function and less physical decline [15]. Factors that reduce the range of activities, such as poor access to safe transportation, also inhibit social participation [16]. Older adults with good social networks, including many close friends and associates and invitations to go out and participate in activities, are more likely to participate in social activities [17]. This suggests a relationship between social contact and social space

mobility. It is also likely that social networks affect social contact. Older people with good social participation have also been found to be more likely to strengthen their emotional ties with others [18]. Therefore, social contact that expands social space mobility and social support relationships can increase social participation and lead to a healthy and longer life among third agers who are living with changes in their social space and social relations owing to aging. In other words, such social contact can promote health among third agers, enrich their personal lives, and extend their healthy life expectancy. Thus, it is important to promote social contact among third agers.

Self-efficacy is an important concept in enabling behaviors that promote social contact among third agers. Bandura defined self-efficacy as individual belief in personal ability to succeed in specific situations or accomplish a task, and stated that it affects behavior [19]. Based on the premise that preventive actions by individuals are effective for healthy longevity of older adult, it is possible to increase self-efficacy for social contact among third agers, which will enable them to take actions to identify, maintain, and expand social contacts. Social contact self-efficacy in this study is defined as the ability of third agers to act toward achieving desirable levels of social participation. Improving social contact self-efficacy in third agers can also support improved physical functioning and mental health among older adults, leading to a more rewarding life and healthier longevity. However, so far, no scales have been developed to evaluate and support this social contact among third agers. To evaluate and support the social contact of third agers, it is essential to develop a scale that allows us to understand and support their social contact. There are two reasons that demonstrate the importance of developing a measure specifically for third agers. First, health promotion in third agers will lead to prevention of the need for health and nursing care in later years, thus extending the healthy life expectancy of these individuals. Second, by promoting the health of third agers, the community can be revitalized to create a healthy community environment for older people. We developed the Social Contact Self-Efficacy Scale for Third Agers (SET) and examined its reliability and validity.

## Methods

### Phase 1: Developing the instrument

First, we developed a pool of items based on the literature review. We searched PubMed, Web of Science, and the Ichushi-web for relevant articles from the perspective of indicators that third agers aim to achieve in order to live a fulfilling third age period. We searched the literature using the following keywords: third age, retirement, elderly, social participation, vitality, reason for living, healthy longevity, and quality of life [20–24]. An item pool was created based on the information obtained from the literature. Item selection criteria were based on two aspects: (1) indicators of enhancement and extension of the third age period; (2) practically beneficial items: the items can be variable across interventions and each item must have clarity of logic, meaning, and readability for ease of understanding among third agers. Using these perspectives, we identified a pool of draft items and made some changes. This resulted in a final list of 34 items.

Second, the pool of items was reviewed by three experts, one researcher, and five older adults. The experts were public health nurses with experience in supporting healthy older adults. The researcher specialized in community health nursing and is studying adaptation to changes in the environment of the elderly after retirement. The older adults were selected from residents of Yokohama City who have not received nursing care certification. They assessed the content validity, face validity, and practical usefulness of the items. Following the reviewers' opinions, we revised the wording of each item. As a result, the modified scale was

reduced to 18 items. Each item was scored on a four-point Likert scale, ranging from 0 (Not confident at all) to 3 (Completely confident).

## Phase 2: Validating the instrument

**Participants and settings.** The survey was conducted with 2,600 community-dwelling elderly aged 65 years and older living in Yokohama City. The subjects were randomly sampled and selected by the Civic Affairs Bureau from the City of Yokohama's Basic Resident Registers. In calculating the sample size, we first assumed a response rate of 30%. This was calculated on the basis of two assumptions indicating the response rate would be slightly lower than that of previous studies. First, the response rate in similarly designed previous studies targeting older people was generally 30%–50%, and second, a certain number of the independent older people targeted in this study were expected to be employed. Next, we assumed that factor analysis would require more than 10 times the number of data items. Furthermore, the design involved conducting exploratory factor analysis and confirmatory factor analysis in separate groups. Because the tentative version of the item pool had 34 items, we calculated 34 items × 10 times × 2 groups × response rate of 30%, resulting in 2,250 respondents. Finally, when randomly selecting the sample from the basic resident register, it was impossible to exclude those who were certified as needing nursing care owing to the nature of the register system; we therefore increased the sample size by 20%, to 2,600 people, assuming that approximately 18% (the rate of certification for needing nursing care in Yokohama City) would be excluded. The inclusion criteria were as follows: (1) age 65 years and over; (2) living in the community (not in a hospital or in residential care); and (3) being an independent older person. The criteria for being an independent older person was identified according to the Certified Level of Need for Long-Term Care National Insurance of Japan (*Kaigo Hoken* in Japanese). Individuals who were not certified as needing long-term care or support were considered independent older people. The exclusion criteria were as follows: (1) less than 65 years old or unknown age; (2) not living in the community (i.e., living in a hospital or in residential care); and (3) having a certified need for long-term care or support. Participants were randomly sampled and selected by the Civic Affairs Bureau of the City of Yokohama's Basic Resident Registers. The participants were recruited and the research took place in Yokohama City, Japan. The City of Yokohama was selected for two reasons. This is because Yokohama is the largest ordinance-designated city in Japan, and the aging of the population is expected to be prominent mainly in urban areas. Yokohama is a city with a population of about 37,000, consisting of 18 administrative districts. The elderly population accounts for 24.6%, of which about 18% are certified as needing long-term care.

The data collection period was September 2020, and the method was to mail explanatory documents (informed consent) and questionnaires to the target people. Participants were considered to be participants by completing the self-administered survey form and returning it to the research institution using a return envelope. A total of 1139 study participants completed the questionnaire, and 978 were considered valid responses, excluding those identified as needing long-term care and those under the age of 65 (filled in by the target's child). The analysis included 790 people with no missing items on the SET or criterion-related validity index items. The basic demographics of the 188 respondents who were excluded owing to missing response data are as follows: the average age was 76.9 years and 59% were women. We believe that the exclusion of the missing data did not cause significant bias in the data. Additionally, the response rate in this study was adequate compared with previous studies among similar older adult populations.

**Measures.** The participants' demographic characteristics included age, sex, working status, frequency of going out, residence years, people living together, disease under treatment

**Table 1. Participants' demographic characteristics.**

| n = 790 | | Number or Mean±SD[a] | % or (Range) |
|---|---|---|---|
| Age (years) | | 73.7±5.8 | (65–93) |
| | 65–74 years old | 472 | 59.7 |
| | 75 years old and older | 307 | 38.9 |
| | Missing | 11 | 1.4 |
| Sex | Female | 407 | 51.5 |
| | Missing | 2 | 0.3 |
| Working employment | Yes | 254 | 32.2 |
| | Missing | 1 | 0.1 |
| Frequency of going out | | 4.4±2.0 | (0–7) |
| | Missing | 8 | 1.0 |
| Residence years | | 31.5±17.4 | (0.3–84) |
| | Missing | 22 | 2.8 |
| People living together | Number | 1.3±0.9 | (0–5) |
| | Yes | 650 | 82.3 |
| Living status | Living with spouse | 565 | 71.5 |
| | Living with children | 259 | 32.8 |
| | Living alone | 138 | 17.5 |
| | Others | 154 | 19.5 |
| | Missing | 2 | 0.3 |
| Disease under treatment | Number | 1.4±1.1 | (0–7) |
| | Yes | 598 | 75.7 |
| Type of disease | High blood pressure | 295 | 37.3 |
| | Visual impairment | 124 | 15.7 |
| | Musculoskeletal diseases | 101 | 12.8 |
| | Diabetes mellitus | 86 | 10.9 |
| | Urinary system disease | 78 | 9.9 |
| | Heart disease | 69 | 8.7 |
| | Others | 280 | 35.4 |
| | Missing | 27 | 3.4 |

[a]SD: standard deviation

(Table 1). Participants responded to 18 items on the modified SET on a four-case Likert scale: 0 = Not confident at all, 1 = Slightly unconfident, 2 = Slightly Confident, 3 = Completely confident. Three measures were used to assess the construct validity of the SET. The first was the General Self-Efficacy Scale (GSES) [25], a measure of an individual's high and low general self-efficacy perceptions, and a higher score indicating greater self-efficacy. The second was the Japan Science and Technology Agency Index of Competence (JST-IC) [26], which measures the ability of older people to perform various activities. Higher scores indicate greater activity ability. The third scale was for subjective health status, and was a single question about perceptions of health answered on a four-point scale. Responses are: 1 = Very healthy, 2 = Quite healthy, 3 = Not very healthy, 4 = Not at all healthy. The score was reversed with SPSS, and was used in four stages, where a higher score showed better health.

**Statistical analyses.** All analyses used IBM SPSS Statistics 25.0 and Amos 25.0 (Chicago, Illinois, USA). Item analysis was used to investigate the reliability of the scale and exploratory factor analysis to investigate the factor structure of the scale. Exclusion criteria for item

analysis included distribution ("Slightly confident" and "Completely confident" were over 90%; kurtosis and skewness were over ± 1.0), rates of response difficulty (non-respondents ≥ 5%), correlations between items (correlation coefficient > 0.6), item–total analysis (correlation coefficient r ≥ 0.6 or p < 0.05), and good–poor analysis (no significant differences between the highest- and lowest-scoring groups).

We randomly divided the total sample (n = 790) into two sub-samples for cross-validation: group 1 (n = 395) for exploratory factor analysis and group 2 (n = 395) for confirmatory factor analysis. We examined the items remaining after item analysis using exploratory factor analysis (maximum likelihood method) with promax rotation [27]. Using the eigenvalues and scree plots, we estimated that there were two factors. We then repeated the exploratory factor analysis, assuming two factors and excluding items with item loadings < 0.4. We determined factor reliability by a Cronbach's alpha ≥ 0.7, and construct validity was verified with confirmatory factor analysis. We examined model fit using the goodness-of-fit index (GFI), adjusted GFI (AGFI), comparative fit index (CFI), and root mean square error of approximation (RMSEA). The model was accepted if the GFI and AGFI were ≥ 0.90, CFI was ≥ 0.95, and RMSEA was ≤ 0.05 [28]. We also examined criterion-related validity by correlating the SET total score with the GSES and JST-IC total score and subjective health perception. We evaluated a correlation of ≥ 0.50 as adequate.

**Ethical considerations.**   This study was conducted with the approval of the Institutional Review Board of the Medical Department of Yokohama City University School (Approval No. A200700002). All study participants provided written informed consent and completed the questionnaire, which was unsigned to ensure their anonymity. The informed consent form explained the voluntary nature of participation, management of data, and intention to publish the results.

## Results

### Demographic characteristics

Table 1 shows the demographic characteristics of the participants. The mean age was 73.7 years. In all, 51.5% were female, 32.2% were employed (including full-time and part-time), 82.3% lived with their spouse, children or others, and 75.7% had a medical condition currently being treated. On average, participants went out 4.4 days/week, and the average number of years of residence was 31.5 years. Laslett, who proposed the definition of "third age", pointed out that third agers are free from social obligations, but he did not necessarily refer to whether they may be employed. In previous studies, third agers were included without distinguishing between those who were working and those who were not; therefore, we decided to include third agers who were working in this study.

### Item analysis

Table 2 shows the item analysis results. One item (item 4) met the exclusion criteria for population distribution, one item (item 16) met the exclusion criteria for kurtosis and skewness and nine items (items 6, 7, 11–15, 17, and 18) met the exclusion criteria for inter-item correlation. However, items 7, 11, 13 and 18 were retained. First, we compared items 6 and 7, and items 17 and 18, based on the correlation, and decided to keep items 7 and 18. These two items had higher item–total correlations than items 6 and 17 and were considered to be significant items on the scale. Next, we compared items 11 to 15 based on the correlation. We decided to keep items 11 and 13 because item 13 was less difficult than the correlated items 12 and 15, and item 11 was considered more familiar to third agers than item 14, and was considered to be a significant item on the scale. Seven items (items 4, 6, 12, 14–17) were therefore excluded and 11 items (items 1–3, 5, 7–11, 13 and 18) were retained for the factor analysis.

**Table 2. Item analysis of the "social contact self-efficacy scale for third agers".**

**n = 790**

| No | Item | Population distribution (%)a | Kurtosis b | Skewness b | Item Difficulty c | Inter-item correlation d | Item-total correlation e | Good-poor analysis f | Exclusion |
|---|---|---|---|---|---|---|---|---|---|
| 1 | I can try to go out as much as possible to avoid stay withdrawn. | 89.6 | 0.139 | -0.761 | 2.6 | - | .604** | .000** | |
| 2 | I can find a little enjoyment in everyday life. | 88.7 | -0.189 | -0.470 | 2.4 | - | .628** | .000** | |
| 3 | I can find a relaxing place in a familiar community. | 79.9 | -0.007 | -0.489 | 3.3 | - | .611** | .000** | |
| 4 | I can go shopping for groceries and daily necessities independently. | 95.3 | 4.854 | -2.170 | 2.5 | - | .422** | .000** | × |
| 5 | I can use facilities and public services that are useful for my health. | 72.0 | -0.649 | -0.445 | 3.2 | - | .632** | .000** | |
| 6 | I can act to gain new knowledge and learn something new. | 75.3 | -0.579 | -0.486 | 2.7 | + | .712** | .000** | × |
| 7 | I can try new things positively. | 63.4 | -0.773 | -0.055 | 2.4 | + | .724** | .000** | |
| 8 | I can notice even slight changes in my health. | 88.5 | -0.181 | -0.338 | 2.2 | - | .514** | .000** | |
| 9 | I can easily consult my doctor or specialist about health concerns. | 78.9 | -0.522 | -0.473 | 2.9 | - | .554** | .000** | |
| 10 | I can reach out to person in need on the streets. | 70.5 | -0.621 | -0.100 | 2.2 | - | .582** | .000** | |
| 11 | I can use my experience and abilities to help others. | 64.3 | -0.583 | -0.142 | 2.9 | + | .692** | .000** | |
| 12 | I can participate in voluntary groups and group activities in the community. | 54.2 | -0.681 | -0.008 | 2.6 | + | .636** | .000** | × |
| 13 | I can enjoy interacting with people in everyday life. | 74.2 | -0.381 | -0.353 | 2.4 | + | .708** | .000** | |
| 14 | I can take a responsible role in a group or organization. | 53.5 | -0.616 | -0.019 | 2.4 | + | .698** | .000** | × |
| 15 | I can expand my social circle from friends and acquaintances. | 55.4 | -0.539 | -0.013 | 2.8 | + | .739** | .000** | × |
| 16 | I can use the internet to communicate with my family and people around me. | 57.0 | -1.240 | -0.192 | 2.5 | - | .425** | .000** | × |
| 17 | I can tell my wishes for medical treatment and care at the end of life. | 81.3 | -0.145 | -0.661 | 2.5 | + | .557** | .000** | × |
| 18 | I am able to support each other with my family and others in times of need. | 84.8 | -0.072 | -0.579 | 2.0 | + | .631** | .000** | |

**: P<0.001

Exclusion criteria of the item analyses

a: Percentage of 'Slightly Confident' and 'Completely confident' were over 90% of the sample.

b: Kurtosis and skewness were over ±1.0 of the sample.

c: Percentage of non-respondents is over 5% of the sample which excluding those who were certified as needing long-term care and those under 65 years of age.(n = 978)

d: Correlation between each item is over 0.6.(Rounded to the second decimal place)

e: Correlation coefficient between the item and the total of all the items (but with exception of the item) is over 0.6 or p<0.05

f: Difference of the average score between the highest- and lowest-scoring groups is not significant difference (p≥0.05)

**Table 3. Exploratory factor analysis of the "social contact self-efficacy scale for third agers".**

| No. | Item | Factor I | Factor II | Total scale communality |
|---|---|---|---|---|
| | | social space mobility | social support relationship | |
| 1 | I can try to go out as much as possible to avoid stay withdrawn. | 0.77 | -0.06 | 0.53 |
| 3 | I can find a relaxing place in a familiar community. | 0.71 | 0.03 | 0.58 |
| 2 | I can find a little enjoyment in everyday life. | 0.68 | 0.12 | 0.53 |
| 5 | I can use facilities and public services that are useful for my health. | 0.62 | 0.01 | 0.39 |
| 9 | I can easily consult my doctor or specialist about health concerns. | -0.07 | 0.76 | 0.44 |
| 8 | I can notice even slight changes in my health. | -0.02 | 0.68 | 0.51 |
| 18 | I am able to support each other with my family and others in times of need. | 0.09 | 0.58 | 0.37 |
| 10 | I can reach out to person in need on the streets. | 0.16 | 0.49 | 0.41 |
| Cronbach's alpha | | 0.80 | 0.75 | 0.83 |
| Cumulative contribution (%) | | 40.3 | 47.1 | |
| Factor correlation coefficients (r) | Factor I | 1.00 | | |
| | Factor II | 0.67 | 1.00 | |

Maximum likelihood solution method with promax rotation.

## Factor structure

The results of the exploratory factor analysis are shown in Table 3. The eigenvalues were 5.145 for one factor, 1.092 for two factors and 0.841 for three factors. The eigenvalues and scree plots suggested a two-factor model. We repeated the exploratory factor analysis with promax rotation until the factor loadings exceeded 0.4. We then excluded items 7, 11 and 13 because the factor loading did not exceed 0.4 in any analysis. Differences in factor loadings between each factor became apparent, allowing the factors to be explained theoretically. Excluding items with loadings of less than 0.4 yielded a two-factor solution. For the final version of the scale, eight items were extracted from the two factors. Factor 1 contained four items (items 1, 2, 3, and 5) and could be interpreted as "social space mobility". This is the self-efficacy to expand that social space during the third age period. Factor 2 included four items (items 8, 9, 10 and 18) and can be interpreted as "social support relationship". This is the self-efficacy to build a social support relationship in the third age period. Factor loadings were above 0.4 for each factor, and the cumulative contribution of the two factors explained 47.1% of the variance. The correlation coefficient between the two factors was 0.67 (Table 3).

## Internal consistency and validity

Two factors were entered as latent factors in the confirmatory factor analysis model. The model fit showed GFI = 0.976, AGFI = 0.955, CFI = 0.982, and RMSEA = 0.050. These results

n=395

**Fig 1. Confirmatory factor analysis of the "social contact self-efficacy Scale for Third Agers (SET)".**

met the appropriate criteria for all subjects (Fig 1). Construct validity was therefore demonstrated. The Cronbach's alpha coefficients were 0.80 for factor 1, 0.75 for factor 2, and 0.83 for the entire scale.

Pearson's correlation analysis showed a correlation between the total SET score and the GSES, JST-IC, and subjective health status. The SET showed a high positive correlation with the GSES (r = 0.552, p < 0.001) and JST-IC (r = 0.495, p < 0.001) and a moderate positive correlation with subjective health status (r = 0.361, p < 0.001) (Table 4).

## Discussion

The SET shows sufficient reliability and validity for a scale of social contact self-efficacy for third agers in Japan. The originality of this scale is that it focuses on self-efficacy for social contact. The previous scales associated with social contacts of older people have been limited to quantitative single measurements, such as those that measure the scope of daily life-space [29],

**Table 4. Criteria-related validity of "the social contact self-efficacy Scale for Third Agers (SET)".**

| n = 790 | | | | |
|---|---|---|---|---|
| Factors | Mean (SD) | GSES | JST-IC | Subjective health |
| I:social space mobility | 8.7(2.3) | .507** | .451** | .378** |
| II:social support relationship | 8.4(2.2) | .477** | .433** | .263** |
| Total 8 items | 17.1(4.1) | .552** | .495** | .361** |

Pearson's correlation coefficients between the total score of validity measure of the SET

SD: standard deviation

GSES: the General Self-Efficacy Scale

JST-IC:the Japan Science and Technology Agency Index of Competence

**: p<0.001

and the number of people who can provide social support when necessary [30]. The SET is also original in that it can be measured in terms of both the social space mobility and social support relationship of third agers.

The first element of the SET is items reflecting self-efficacy for social space mobility, and a high score for this element indicates that older adults are highly motivated to expand the scope of their activities. Third agers are identified as having a need to acquire new places to go as a result of changes in their social activities. Decreased mobility of living spaces is known to have a negative impact on health, and previous studies have found associations with falls, fractures [31], physical functional vulnerability [32], mortality [33, 34] and subjective health status [35]. In summary, measuring self-efficacy for expanding social space mobility will lead to finding new places to go, enjoyment, and resources useful for health in individuals' immediate surroundings, and to going out. Therefore meaningful both in the third age, and in preventing the onset of the fourth age for as long as possible.

The second component of the SET reflects self-efficacy for social support relationships. High scores on this factor suggest high levels of motivation to expand the social networks. Third agers have a need to develop new relationships that allow them to have a sense of belonging and responsibility. For older people, an expanded social network improves quality of life [36, 37] and facilitates access to social support [38–40]. It has also been suggested that poor social networks are associated with mental illness, and that socially disconnected people are more likely to commit suicide than others [41]. Measuring self-efficacy for social support relationship expansion can lead to having opportunities to think about one's own health so as to maintain relationships with others, and to expand new relationships that involve "caring for each other", which could be important when considering the transition from the third to the fourth age.

The clinical usefulness of the scale is threefold. First, the SET can help third agers to understand and expand their social contacts after retirement. Second, it can help professionals to assess individuals' social contact and provide personalized information and advice on social participation. Third, the SET can also provide useful information for building a system and community in which third agers are more likely to make social contact. In other words, this scale can be useful in motivating individuals and building the community, and as a result, the people involved can improve their social contacts within society. In the future, effective programs and systems can disseminate the SET to help promote social contact opportunities for third agers and extend the third age phase, helping older adults to maintain physical functioning and improve their quality of life.

## Limitation

This study had a few limitations. First, it was cross-sectional and its predictive validity is unknown. Longitudinal studies are needed to determine the degree to which SET scores are related to social contact in the future. Second, although the study setting was a major city in Japan, according to previous research [42], social contact beliefs and behaviors among third agers may vary by culture and resources in different regions or countries. Further studies are needed to investigate the construct validity of the SET in other regions and countries that may have different characteristics from the present participants.

## Supporting information

**S1 Fig. Appendix.** The SET English version.
(TIF)

**S2 Fig. Appendix.** The SET Japanese version.
(TIF)

**S1 Dataset. Anonymized minimal data set.**
(CSV)

# Acknowledgments

We thank Associate Professor Azusa Arimoto, Assistant Professors Kae Shiratani, Eriko Ito, and all members of the Department of Community Health Nursing, Graduate School of Medicine, Yokohama City University. Most of all, we thank all the third agers and experts who graciously gave their time and energy to participate in this study. We also thank Analisa Avila from Edanz Group (https://www.jp.edanz.com/ac) for editing a draft of this manuscript.

# Author Contributions

**Conceptualization:** Moemi Oki, Etsuko Tadaka.

**Data curation:** Moemi Oki.

**Formal analysis:** Moemi Oki.

**Funding acquisition:** Etsuko Tadaka.

**Investigation:** Moemi Oki.

**Methodology:** Moemi Oki.

**Project administration:** Etsuko Tadaka.

**Resources:** Etsuko Tadaka.

**Software:** Moemi Oki.

**Supervision:** Etsuko Tadaka.

**Validation:** Moemi Oki, Etsuko Tadaka.

**Visualization:** Moemi Oki.

**Writing – original draft:** Moemi Oki.

**Writing – review & editing:** Moemi Oki, Etsuko Tadaka.

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
