## [Decision Letter · Decision Letter 0]

1 Apr 2021

PONE-D-21-01370

Development of a social contact self-efficacy scale for ‘third agers’ in Japan

PLOS ONE

Dear Dr. Tadaka,

Thank you for submitting your manuscript to PLOS ONE. After careful consideration, we feel that it has merit but does not fully meet PLOS ONE’s publication criteria as it currently stands. Therefore, we invite you to submit a revised version of the manuscript that addresses the points raised during the review process.

Several major revisions are needed in the present form. 

See the Reviewer’s comments and respond them appropriately.

We look forward to receiving your revised manuscript.

Kind regards,

Masaki Mogi

Academic Editor

PLOS ONE

Journal Requirements:

In your Methods section, please provide a justification for the sample size used in your study, including any relevant power calculations (if applicable).

In your Methods section, please provide additional information about the participant recruitment method and the demographic details of your participants. Please ensure you have provided sufficient details to replicate the analyses such as: a) a description of any inclusion/exclusion criteria that were applied to participant recruitment, b) a description of how participants were recruited, and c) descriptions of where participants were recruited and where the research took place.

In your Data Availability statement, you have not specified where the minimal data set underlying the results described in your manuscript can be found. PLOS defines a study's minimal data set as the underlying data used to reach the conclusions drawn in the manuscript and any additional data required to replicate the reported study findings in their entirety. All PLOS journals require that the minimal data set be made fully available. For more information about our data policy, please see http://journals.plos.org/plosone/s/data-availability.

Thank you for stating the following financial disclosure:

4a)         Please clarify the sources of funding (financial or material support) for your study. List the grants or organizations that supported your study, including funding received from your institution.

4b)         State what role the funders took in the study. If the funders had no role in your study, please state: “The funders had no role in study design, data collection and analysis, decision to publish, or preparation of the manuscript.”

4c)          If any authors received a salary from any of your funders, please state which authors and which funders.

4d)         If you did not receive any funding for this study, please state: “The authors received no specific funding for this work.”

Reviewers' comments:

Reviewer's Responses to Questions

**Comments to the Author**

1. Is the manuscript technically sound, and do the data support the conclusions?

Reviewer #1: Partly

2. Has the statistical analysis been performed appropriately and rigorously? 

Reviewer #1: Yes

3. Have the authors made all data underlying the findings in their manuscript fully available?

Reviewer #1: Yes

4. Is the manuscript presented in an intelligible fashion and written in standard English?

Reviewer #1: Yes

5. Review Comments to the Author

Reviewer #1: Thank you for the opportunity to review the manuscript reporting the development of a social contact self-efficacy scale for Japanese "third agers." The manuscript is well-written, and the topic is interesting. Nonetheless, I found the theoretical reasoning for the construct "social contact self-efficacy" is underdeveloped in the current manuscript. The scale items' face validity looks somewhat questionable to me, which could hinder the adoption of this scale in future studies. The following comments and suggestions are for the authors’ considerations.

1. Why is it important to develop a measurement specifically for the third agers? How's this group's social contact similar to and different from people in other age groups?

2. Among the 2,600 potential participants, 1,139 responded. Is there any risk of self-selection bias? For example, 82.3% of the study sample lived with a spouse, children, or others? Is this rate expected for the older adult population in Yokohama city?

3. In the conclusion section of the abstract, the authors mentioned, "this scale may help third agers in gaining and expanding opportunities for social contact." How can a scale help third agers in gaining contact?

4. The authors proposed two possible elements in social relationships, i.e., social space mobility and social support network. Although some definitions of the two concepts were provided in lines 58-60, I still found myself not so clear about these two aspects' meaning. Is social space mobility related to living space and mobility or purely how diverse one's social network is? Does social support network sound like the frequency of contact with people in the network? The social support network is a widely used term in social relationship research. It is a nuanced concept that could include network size, frequency of contact, and relationships with others in the network. The definitions here seem not clear enough.

5. The proposed two possible elements in social relationships are intriguing. However, what is the theoretical rationale that supports this taxonomy? How are these two aspects related to self-efficacy? Any reference to support the legitimacy of this taxonomy?

6. Consider defining the concept "social contact self-efficacy."

7. Line 96, the author mentioned "practically beneficial" as a criterion for selecting scale items. Could you please elaborate what are the practical benefits that were considered?

8. Line 128, what’s recovery rate?

9. The face validity of the scale items shown in table 3 could be somewhat questionable. For example, Item 9, "I can easily consult my doctor or specialist about health concerns," and item 8, "I can notice even slight changes in my health." How do these two items related to social contact self-efficacy?

10. Line 272, the authors argued that a higher score on the social support network dimension indicates higher motivation to expand the social network. I am not sure if items 8 and 9 could reflect that.

Editing suggestions

11. Line 111, the survey was conducted “with” 2,600 community-dwelling older adults instead of “on”.

12. Line 125, here the deficiencies mean missingness?

13. “The basic attributes of the 188 individuals who were excluded owing to missing data were mean age 76.9 years and 59% women." This sentence appears awkward. Please rephrase it.

6. PLOS authors have the option to publish the peer review history of their article (what does this mean?). If published, this will include your full peer review and any attached files.

Reviewer #1: No

---

## [Author Response · Author response to Decision Letter 0]

15 May 2021

Response letter

Dr. Masaki Mogi

Academic Editor

PLOS ONE 

12 May 2021

Dear Dr. Masaki Mogi,

Thank you very much for your e-mail regarding our manuscript, “Development of a social contact self-efficacy scale for ‘third agers’ in Japan” (PONE-D-21-01370). We are delighted to hear that it is potentially acceptable for publication in PLOS ONE. Please find attached a revised version of our manuscript.

Your comments and those of the reviewers were highly insightful and enabled us to greatly improve the quality of our manuscript. Below, we present our point-by-point responses to each of the comments made by the reviewer. 

We look forward to hearing from you regarding our resubmission. We would be glad to respond to any further questions and comments that you may have.

In closing, the contact information for Etsuko Tadaka, Principal Investigator, has changed since April. The new contact information is below, and the manuscript has been revised as well. Thank you very much in advance for your kind support.

E-mail: e_tadaka@pop.med.hokudai.ac.jp (ET)

Sincerely yours,

Moemi Oki, MSN, RN, PHN

Department of Community Health Nursing, Graduate School of Medicine, Yokohama City University, Yokohama, Kanagawa, Japan

Tel: +81 80 1156 2407 E-mail: mopmick0504@gmail.com

and https://journals.plos.org/plosone/s/file?id=ba62/PLOSOne_formatting_sample_title_authors_affiliations.pdf

Author’s Response:

We have checked PLOS ONE's style requirements and modified the manuscript accordingly. Thank you for your comment.

2) In your Methods section, please provide a justification for the sample size used in your study, including any relevant power calculations (if applicable).

Author’s Response:

In our Methods section, we provided the justification for the sample size used in the study, including relevant power calculations.

Page9, Lines 127, Methods

In calculating the sample size, we first assumed a response rate of 30%. This was calculated on the basis of two assumptions indicating the response rate would be slightly lower than that of previous studies. First, the response rate in similarly designed previous studies targeting older people was generally 30%–50%, and second, a certain number of the independent older people targeted in this study were expected to be employed. Next, we assumed that factor analysis would require more than 10 times the number of data items. Furthermore, the design involved conducting exploratory factor analysis and confirmatory factor analysis in separate groups. Because the tentative version of the item pool had 34 items, we calculated 34 items × 10 times × 2 groups × response rate of 30%, resulting in 2,250 respondents. Finally, when randomly selecting the sample from the basic resident register, it was impossible to exclude those who were certified as needing nursing care owing to the nature of the register system; we therefore increased the sample size by 20%, to 2,600 people, assuming that approximately 18% (the rate of certification for needing nursing care in Yokohama City) would be excluded.

In your Methods section, please provide additional information about the participant recruitment method and the demographic details of your participants. Please ensure you have provided sufficient details to replicate the analyses such as:

 a) a description of any inclusion/exclusion criteria that were applied to participant recruitment, b) a description of how participants were recruited, and c) descriptions of where participants were recruited and where the research took place.

Author’s Response:

The inclusion criteria were as follows: (1) age 65 years and over; (2) living in the community (not in a hospital or in residential care); and (3) being an independent older person. The criteria for being an independent older person was identified according to the Certified Level of Need for Long-Term Care National Insurance of Japan (Kaigo Hoken in Japanese). Individuals who were not certified as needing long-term care or support were considered independent older people. The exclusion criteria were as follows: (1) less than 65 years old or unknown age; (2) not living in the community (i.e., living in a hospital or in residential care); and (3) having a certified need for long-term care or support. Participants were randomly sampled and selected by the Civic Affairs Bureau of the City of Yokohama's Basic Resident Registers. The participants were recruited and the research took place in Yokohama City, Japan.

Page9, Lines 140, Methods

The inclusion criteria were as follows: (1) age 65 years and over; (2) living in the community (not in a hospital or in residential care); and (3) being an independent older person. The criteria for being an independent older person was identified according to the Certified Level of Need for Long-Term Care National Insurance of Japan (Kaigo Hoken in Japanese). Individuals who were not certified as needing long-term care or support were considered independent older people. The exclusion criteria were as follows: (1) less than 65 years old or unknown age; (2) not living in the community (i.e., living in a hospital or in residential care); and (3) having a certified need for long-term care or support. Participants were randomly sampled and selected by the Civic Affairs Bureau of the City of Yokohama's Basic Resident Registers. The participants were recruited and the research took place in Yokohama City, Japan.

3) In your Data Availability statement, you have not specified where the minimal data set underlying the results described in your manuscript can be found. PLOS defines a study's minimal data set as the underlying data used to reach the conclusions drawn in the manuscript and any additional data required to replicate the reported study findings in their entirety. All PLOS journals require that the minimal data set be made fully available. For more information about our data policy, please see http://journals.plos.org/plosone/s/data-availability. 

Author’s Response:

We will update your Data Availability statement to reflect the information you provide in your cover letter. In our Data Availability statement, we described this as follows:

Data Availability:

All relevant data are within the manuscript and its Supporting Information files. 

S1 Fig. Appendix. The SET English version.

S2 Fig. Appendix. The SET Japanese version.

S3 Anonymized minimal data set

4) Thank you for stating the following financial disclosure:

4a) Please clarify the sources of funding (financial or material support) for your study. List the grants or organizations that supported your study, including funding received from your institution.

4b) State what role the funders took in the study. If the funders had no role in your study, please state: “The funders had no role in study design, data collection and analysis, decision to publish, or preparation of the manuscript.”

4c) If any authors received a salary from any of your funders, please state which authors and which funders.

4d) If you did not receive any funding for this study, please state: “The authors received no specific funding for this work.” Please include your amended statements within your cover letter; we will change the online submission form on your behalf. 

Author’s Response:

4a) The sources of funding for our study were Grants-in-Aid for Scientific Research (KAKENHI) of the Japan Society for the Promotion of Science (PI: Dr. Etsuko TADAKA).

4b) The funders had no role in study design, data collection and analysis, decision to publish, or preparation of the manuscript.

4c) No authors received a salary or fees from any of the funders.

Comments of Reviewer #1

1) Why is it important to develop a measurement specifically for the third agers? How's this group's social contact similar to and different from people in other age groups? Why is it important to develop a measurement specifically for the third agers? 

Author’s Response:

There are two reasons that demonstrate the importance of developing a measure specifically for third agers. First, health promotion in third agers will lead to prevention of the need for health and nursing care in later years, thus extending the healthy life expectancy of these individuals. Second, by promoting the health of third agers, the community can be revitalized to create a healthy community environment for older people.

How's this group's social contact similar to and different from people in other age groups?

One similarity is the importance of social contact, which is the interaction that takes place between an individual and members of society and is a component of society. One difference is the process by which social contacts are formed. Third agers are thought to form relationships with a relatively limited extent of people, and in limited locations outside of the workplace, in their retired lives. People of other ages. such as “second agers”, are thought to form relationships with a relatively wide range of people and in more locations, such as at school and work, during their active lives.

We have added the following to the introduction section.

Page7, Lines 92, Introduction

There are two reasons that demonstrate the importance of developing a measure specifically for third agers. First, health promotion in third agers will lead to prevention of the need for health and nursing care in later years, thus extending the healthy life expectancy of these individuals. Second, by promoting the health of third agers, the community can be revitalized to create a healthy community environment for older people.

2) Among the 2,600 potential participants, 1,139 responded. Is there any risk of self-selection bias? For example, 82.3% of the study sample lived with a spouse, children, or others? Is this rate expected for the older adult population in Yokohama city? 

Author’s Response:

As you pointed out, there is certainly the possible risk of self-selection bias; however, we believe that a certain level of representativeness is ensured in our population. In the latest survey among older people conducted by Yokohama City (where the present research was conducted), 81.1% of respondents lived with a spouse, children, or others. This is consistent with the results of this survey, and this rate is expected for the older adult population in Yokohama City. Additionally, in this study, the target population was randomly sampled and the collection rate was sufficient in comparison with previous studies. We therefore believe that the survey is representative.

3) In the conclusion section of the abstract, the authors mentioned, "this scale may help third agers in gaining and expanding opportunities for social contact." How can a scale help third agers in gaining contact? 

Author’s Response:

The present scale can help third agers to build their social contacts in the following ways.

First, the scale will allow third agers to understand and reflect on their social contacts, which can help them to expand their social activities and relationships for the rest of their lives. Second, this scale can provide useful information for community organizations and public health policy makers for the development of systems and communities that contribute to people's social contacts. This means not only improving the motivation of third agers themselves but also promoting the above two aspects of community building; as a result, the people involved can improve their contacts within society.

We have added the following to the discussion section.

Page22, Lines 325, Discussion

In other words, this scale can be useful in motivating individuals and building the community, and as a result, the people involved can improve their social contacts within society. In the future, effective programs and systems can disseminate the SET to help promote social contact opportunities for third agers and extend the third age phase, helping older adults to maintain physical functioning and improve their quality of life.

4) The authors proposed two possible elements in social relationships, i.e., social space mobility and social support network. Although some definitions of the two concepts were provided in lines 58-60, I still found myself not so clear about these two aspects' meaning. Is social space mobility related to living space and mobility or purely how diverse one's social network is? Does social support network sound like the frequency of contact with people in the network? The social support network is a widely used term in social relationship research. It is a nuanced concept that could include network size, frequency of contact, and relationships with others in the network. The definitions here seem not clear enough. 

Author’s Response:

We have reconsidered the two elements in social relationships based on your valuable comments. As a result, social space mobility has been modified to social space mobility and social support network to social support relationships. We redefined the first element, social space mobility, as related to both living space and mobility. We also included both physical space (outdoors) as well as cyberspace or virtual space (online) within the scope of social space mobility. Next, social support network was defined as not only the frequency of contact with other people but also the ability to build and maintain new relationships (including with friends, neighbors, public and private support agencies, and primary care physician) while respecting the existing relationships during individuals’ retired lives; we have modified the second element to social support relationship.

We have added the following to the introduction section.

Page5, Lines 58, Introduction

There are two elements in social relationships that contribute to improved health and longevity among third agers, namely, “social space mobility” and “social support relationships”. “Social space mobility” is defined as extent of individuals’ social contacts, as related to both living space and mobility. “Social support relationships” is defined as interactions with individuals’ social contacts as well as the ability to build and maintain new social relationships (including with friends, neighbors, public and private support agencies, and primary care physicians) while preserving existing relationships in one’s retired life. Enhanced social participation among older people means maintaining wide social space mobility and enhanced social support relationships.

5) The proposed two possible elements in social relationships are intriguing. However, what is the theoretical rationale that supports this taxonomy? How are these two aspects related to self-efficacy? Any reference to support the legitimacy of this taxonomy? 

Author’s Response:

Regarding the theoretical rationale supporting this taxonomy, the reason we focused on these two elements in this study is because it has recently been reported that the mortality rate among elderly people with overlapping conditions of both "social isolation" (non-interaction) and "confinement" (not going outside of the home) is significantly higher than that of their counterparts with only one of these conditions (Sakurai et al., 2019). To effectively prevent both social isolation and confinement among older people in a country where the shortage of human resources is expected to become more serious with a decline in the population, it is necessary to take measures that focus on both risk factors at the same time. 

Regarding how are these two aspects are related to self-efficacy, we focused on self-efficacy for two reasons. First, we used self-efficacy because it has been theoretically shown that self-efficacy promotes more desirable behaviors. Second, the measures of social participation for the elderly that have been developed to date have been limited to quantitative measures, such as measuring the types of community activities in which individuals participate, the frequency of outings, and the number of people in their social support network.

Refernce) 

Sakurai R, Yasunaga M, Nishi M, Fukaya T, Hasebe M, Murayama Y, et al. Co-existence of social isolation and homebound status increase the risk of all-cause mortality. Int Psychogeriatrics. 2019;31: 703–711. doi:10.1017/S1041610218001047.

6) Consider defining the concept "social contact self-efficacy." 

Author’s Response:

Social contact self-efficacy in this study is defined as the ability of third agers to act toward achieving desirable levels of social participation, which is particularly relevant in the two major domains, social space mobility and social support relationship.

We have added the following to the introduction section.

Page6, Lines 85, Introduction

Social contact self-efficacy in this study is defined as the ability of third agers to act toward achieving desirable levels of social participation. Improving social contact self-efficacy in third agers can also support improved physical functioning and mental health among older adults, leading to a more rewarding life and healthier longevity.

7) Line 96, the author mentioned "practically beneficial" as a criterion for selecting scale items. Could you please elaborate what are the practical benefits that were considered? 

Author’s Response:

We considered items to be "practically beneficial" as follows; the items can be variable across interventions; and the items have clarity of logic, meaning, and readability for ease of understanding among third agers.

We have added the following to the methods section.

Page7, Lines 107, Methods

Item selection criteria were based on two aspects: (1) indicators of enhancement and extension of the third age period; (2) practically beneficial items: the items can be variable across interventions and each item must have clarity of logic, meaning, and readability for ease of understanding among third agers.

8) Line 128, what’s recovery rate? 

Author’s Response:

The recovery rate has been corrected to the response rate. We were able to collect 1,139 of 2,600 questionnaires, so the response rate was calculated to be 43.8%.

We have added the following to the methods section.

Page11, Lines 166, Methods

Additionally, the response rate in this study was adequate compared with previous studies among similar older adult populations.

9) The face validity of the scale items shown in table 3 could be somewhat questionable. For example, Item 9, "I can easily consult my doctor or specialist about health concerns," and item 8, "I can notice even slight changes in my health." How do these two items related to social contact self-efficacy? 

Author’s Response:

The face validity of the scale items, i.e., how items 8 and 9 relate to social contact self-efficacy, are discussed below one item at a time.

Item 9: "I can easily consult my doctor or specialist about health concerns" is meant to assess the ability to stay connected with medical care. Even among third agers with high levels of good health, medical needs are expected to increase as they age. These individuals need to be prepared to quickly connect with medical care that is appropriate for them when they need it; through consultations, they can build health-related networks and trusting relationships with health care professionals.

Item 8: "I can notice even slight changes in my health" is meant to assess the ability to pay attention to one's own health so as to maintain relationships with others. Third agers are required to take greater care of themselves because they may gradually lose the means to manage their health when they retire, such as attending regular medical checkups that were provided by their employer. Additionally, with the expected prevalence of infectious diseases and future social climate, it is necessary for older people to be aware of changes in their physical condition to judge whether they can interact with others and to build relationships in which they can care for each other's physical condition.

10) Line 272, the authors argued that a higher score on the social support network dimension indicates higher motivation to expand the social network. I am not sure if items 8 and 9 could reflect that. 

Author’s Response:

Item 8 measures the ability to care for one's own health and to take necessary actions so as to maintain relationships with others. A high score on this item indicates that the individual is aware of the state of their own body during the course of daily activities and is able to care for their own physical condition. For example, being able to care for one another’s physical condition and say "How are you?" to each other when interacting with others, as well as being able to recognize when one's own physical condition is poor and to express it in words, will help in building rewarding relationships for both parties in which older people can spend time together in comfort and at ease. Item 9 measures awareness about taking care of one's own health, selecting appropriate people to turn to in case of an emergency, and the ability to express one's own health condition in one's own words. This is also an item that measures awareness regarding connecting with familiar medical professionals and building trusting relationships. A high score on this item indicates that when individuals feel uncomfortable or worried about their physical condition, they do not leave problems unattended; these individuals have less resistance to visiting a hospital, and they are more likely to take actions that lead to undergoing a medical examination and building connections with medical professionals.

11) Line 111, the survey was conducted “with” 2,600 community-dwelling older adults instead of “on”. 

Author’s Response:

As you pointed out, we have made the following revision:

Page 8, Lines 125, Methods, Phase 2: Validating the instrument, Participants and Settings

“The survey was conducted with 2,600 community-dwelling elderly aged 65 years and older living in Yokohama City.”

12) Line 125, here the deficiencies mean missingness? 

Author’s Response:

As you pointed out, here deficiencies refers to missing items. We made the following revision:

Page 10, Lines 162, Methods, Phase 2: Validating the instrument, Participants and Settings

“The analysis included 790 people with no missing items on the SET or criterion-related validity index items.”

13) “The basic attributes of the 188 individuals who were excluded owing to missing data were mean age 76.9 years and 59% women." This sentence appears awkward. Please rephrase it. 

Author’s Response:

As you pointed out, we have made the following revision:

Page 11, Lines 163, Methods, Phase 2: Validating the instrument, Participants and Settings

“The basic demographics of the 188 respondents who were excluded owing to missing response data are as follows: the average age was 76.9 years and 59% were women. We believe that the exclusion of the missing data did not cause significant bias in the data.”

---

## [Editor Report · Decision Letter 1]

10 Jun 2021

Development of a social contact self-efficacy scale for ‘third agers’ in Japan

PONE-D-21-01370R1

Dear Dr. Tadaka,

We’re pleased to inform you that your manuscript has been judged scientifically suitable for publication and will be formally accepted for publication once it meets all outstanding technical requirements.

Kind regards,

Masaki Mogi

Academic Editor

PLOS ONE

Additional Editor Comments (optional):

The authors well responded to the Reviewer's comments. No further comment.
---

## [Editor Report · Acceptance letter]

14 Jun 2021

PONE-D-21-01370R1 

Development of a social contact self-efficacy scale for ‘third agers’ in Japan 

Dear Dr. Tadaka:

I'm pleased to inform you that your manuscript has been deemed suitable for publication in PLOS ONE. Congratulations! Your manuscript is now with our production department. 

Kind regards, 

on behalf of

Dr. Masaki Mogi 

Academic Editor

PLOS ONE